# Research on Prediction of Surface Deformation in Mining Areas Based on TPE-Optimized Integrated Models and Multi-Temporal InSAR

Sichun Long [1], Maoqi Liu [1,2,3,*], Chaohui Xiong [1], Tao Li [4], Wenhao Wu [1], Hongjun Ding [5], Liya Zhang [1], Chuanguang Zhu [1] and Shide Lu [5]

1 School of Earth Sciences and Spatial Information Engineering, Hunan University of Science and Technology, Xiangtan 411201, China; sclong@hnust.edu.cn (S.L.)
2 School of Resources & Environment and Safety Engineering, Hunan University of Science and Technology, Xiangtan 411201, China
3 Hunan Provincial Key Laboratory of Coal Resources Clean-Utilization of and Mine Environment Protection, Xiangtan 411201, China
4 Satellite Navigation and Positioning Technology Research Centre, Wuhan University, Wuhan 430079, China
5 China Construction Fifth Engineering Division Corp., Ltd., Changsha 410000, China
* Correspondence: mqliu@mail.hnust.edu.cn

**Abstract:** The prevailing research on forecasting surface deformations within mining territories predominantly hinges on parameter-centric numerical models, which manifest constraints concerning applicability and parameter reliability. Although Multi-Temporal InSAR (MT-InSAR) technology furnishes an abundance of data, the underlying information within these data has yet to be fully unearthed. Consequently, this paper advocates a novel methodology for prognosticating mining area surface deformation by integrating ensemble learning with MT-InSAR technology. Initially predicated upon the MT-InSAR monitoring outcomes, the target variables for the ensemble learning dataset were procured by melding distance-based features with spatial autocorrelation theory. In the ensuing phase, spatial stratified sampling alongside mutual information methodologies were deployed to select the features of the dataset. Utilizing the MT-InSAR monitoring data from the Zixing coal mine in Hunan, China, the relationship between fault slippage and coal extraction in the study area was rigorously analyzed using Granger causality tests and Johansen cointegration assays, thereby acquiring the dataset requisite for training the Bagging model. Subsequently, leveraging the Bagging technique, ensemble models were constructed employing Decision Trees, Support Vector Regression, and Multi-layer Perceptron as foundational estimators. Furthermore, the Tree-structured Parzen Estimator (TPE) optimization algorithm was applied to the Bagging model, resulting in an optimal model for predicting fault slip in mining areas. In comparison with the baseline model, the performance increased by 25.88%, confirming the effectiveness of the data preprocessing method outlined in this study. This result also demonstrates the innovation and feasibility of combining ensemble learning with MT-InSAR technology for predicting mining area surface deformation. This investigation is the first to integrate TPE-optimized ensemble models with MT-InSAR technology, offering a new perspective for predicting surface deformation in mining territories and providing valuable insights for further uncovering the hidden information in MT-InSAR monitoring data.

**Keywords:** surface deformation; ensemble learning; multi-temporal InSAR; bagging; causality test

## 1. Introduction

'Fault slip' refers to the sliding phenomenon occurring between fault planes and is categorized into two primary types: one directly induced by internal crustal forces and the other influenced by a combination of internal crustal forces and human activities [1]. Both scenarios can exert profound effects on the Earth's surface topography and morphology

as well as on the safety, economy, and development of human society [2,3]. Research indicates that during the process of coal mining, faults within the mining area may become active, thereby increasing the risk of fault slip and mine disasters. Consequently, studying the faults in mining areas holds significant scientific and practical value for preventing geological disasters and ensuring miner safety [4–7].

Interferometric Synthetic Aperture Radar (InSAR) exemplifies a high-precision, non-contact monitoring technology. It not only boasts the advantage of continuous spatial coverage but also proficiently surveys large-scale ground deformation with an unparalleled density [8]. The advent of Multi-Temporal InSAR (MT-InSAR) has effectively mitigated issues such as temporal and spatial decorrelation as well as atmospheric delays in interferometric measurements [9]. Persistent Scatterer (PS) InSAR, the first generation of multi-temporal technology, continues to be widely applied in urban areas and regions with high coherence. To address the issue of low coherence in some areas, the second-generation MT-InSAR technology, known as 'Distributed Scatterers (DS) InSAR', was developed. Existing research confirms that DS-InSAR is capable of effectively monitoring mining areas with lower coherence [10,11].

In response to the large volume of mining area monitoring data produced by MT-InSAR technology, including ground deformation and fault slip, current research primarily focuses on determining whether deformation has occurred in the study area [12,13]. Additionally, due to the large number of MT-InSAR monitoring points, high data complexity, and interconnections between monitoring points, manually extracting deep insights from these data is a significant challenge [14–16]. Although Machine Learning (ML) has been applied to post-process mining area InSAR data, particularly for classifying InSAR results such as identifying ground deformation, this approach still does not fully meet the comprehensive needs of engineering decision-making [17–20]. Specifically, the needed information goes beyond just detecting current deformation; it includes deeper insights, like forecasting ongoing deformation in the future. Currently, research in this area mainly focuses on parameter-driven numerical models, and some scholars have used basic ML algorithms to enhance existing ground deformation prediction methods. However, these methods are still limited by factors such as parameter reliability, data scarcity, and underutilization of algorithmic performance [21–24]. Overall, considering the limitations of current ground deformation prediction methods in mining areas and the rapid growth of monitoring data, efficiently and comprehensively extracting essential information has become an urgent issue that needs immediate attention.

To address the aforementioned issues, this study employs a Bagging-centric ensemble learning approach for MT-InSAR-derived monitoring data from mining areas. Initially, the ensemble model exhibits strong explanatory power, clearly explaining the specific logic for predicting surface deformation in mining areas, especially in Decision Tree (DT)-based ensemble models. Furthermore, the practical effectiveness of Bagging has been demonstrated in various domains, such as in landslide prediction using Bagging-based ensemble models. These studies highlight that Bagging not only improves the predictive performance of models but also enhances stability by reducing overfitting and increasing generalization capabilities [25–27]. Drawing on these factors, we devised a framework for ground deformation prediction that integrates Bagging with MT-InSAR technology. This framework is divided into data preprocessing and model establishment and aims to efficiently extract in-depth information from complex monitoring data. In the model construction phase, Bagging models based on DT, Support Vector Regression (SVR), or Multi-layer Perceptron (MLP) were constructed. To assess the models' effectiveness, we analyzed MT-InSAR monitoring data from the Zixing coal mine in Hunan Province, China. During data processing, we used the Granger causality test and Johansen cointegration test to investigate the causal relationship between coal mining and fault slip, thereby obtaining the training dataset for the Bagging model. Finally, to enhance the models' performance, we employed the Tree-structured Parzen Estimator (TPE) for optimization, resulting in an optimal model for predicting fault slip in mining areas. Overall, this study offers a new

approach to predicting surface deformation in mining areas and provides a reference for post-processing MT-InSAR data.

## 2. Materials

### 2.1. Study Area

This study focuses on the Zixing coal mine in Hunan Province, China, as shown in Figure 1a. The coal mine serves as a significant economic cornerstone for local development, and the red rectangle in Figure 1b defines the boundaries of the mining expanse. The survey results indicate that the geological structure of this area is primarily monoclinal, oriented northeastward, and exhibits a moderate level of complexity. A fault, approximately 4 km long and indicated by the arrow in Figure 1b, cuts across the entire mining area along its trend and has roads and residential buildings situated along it. Fault slippage poses a significant risk to the safety of the coal mine operations as well as to the lives and property of the residents in the area. Currently, the main causes and trends of this fault movement remain unclear, thereby escalating the risk to the lives and property of residents in the region.

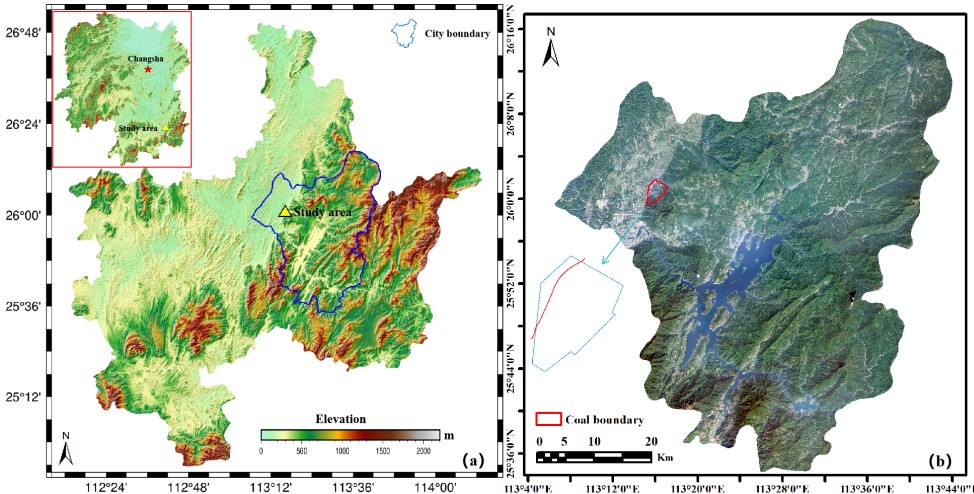

**Figure 1.** Geographic location and overview of the study area. (**a**) Geographic location of the study area. (**b**) Overview of the study area.

### 2.2. Data

The research data were derived from the Sentinel-1A satellite data provided by the European Space Agency; the satellite has a maximum ground subsidence monitoring rate of 42 cm/year. The imagery data captured in the Terrain Observation with Progressive Scans (TOPS) mode have a range resolution of 2.7–3.5 meters and an azimuth resolution of 22 m [10]. This study utilized 162 image scenes from the period between 2017 and 2023. Considering the characteristics of surface deformation in the mining area, these data were divided into six segments, each with a temporal baseline not exceeding 360 days, and each segment was processed individually. Additionally, the small scale of our study area means that the relative error in the vertical direction caused by high-coherence points due to satellite orbital errors is negligible. To reduce errors associated with atmospheric delay, we used data from the Generic Atmospheric Correction Online Service (GACOS) for atmospheric error correction during the processing phase [28,29].

## 3. Methods

To explore the fault movement in the mining area using the Bagging algorithm, this paper proposes utilizing MT-InSAR monitoring results as the dataset for the Bagging model. Initially, we establish the temporal baseline to process long-term SAR data and extract high-coherence points from the study area. Subsequently, employing the spatial autocorrelation

method, we ascertain the range of high-coherence points for the target variables and features in the dataset. Following this, we further employ a distance-based method to extract target variables and use spatial stratified sampling along with a mutual information method for feature selection. Ultimately, we construct the Bagging model and process the obtained target variables and features to derive the complete dataset for training. We also optimize the model using the TPE to attain the optimal ground deformation prediction model. Figure 2 elaborately illustrates the relevant processing procedures.

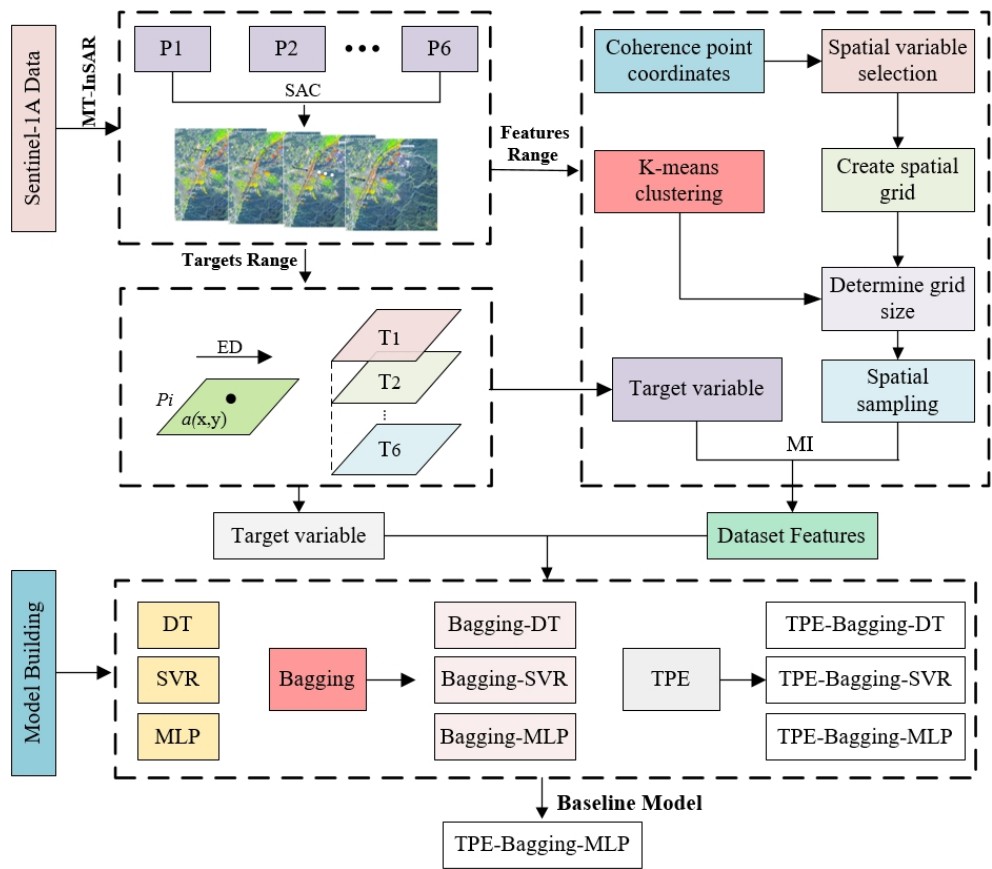

**Figure 2.** Overall methodology flowchart.

### 3.1. MT-InSAR Data Processing

In the TOPS (Terrain Observation by Progressive Scans) mode, the Doppler centroids of adjacent swath images in SAR data change rapidly in the azimuth direction. This rapid change makes the interferometric process between two images particularly sensitive to azimuth registration errors. Therefore, it is essential to keep the azimuth registration error below 0.001 pixels to prevent discontinuities in the interferogram phases. To meet the requirements of interferometric registration of SAR imagery in such scenarios, image registration is initially conducted under precise orbital conditions and is followed by geometric registration. Subsequently, Enhanced Spectral Diversity is utilized to accurately estimate the residual offset post-azimuth geometric registration, thereby achieving an azimuth registration precision of less than 0.001 pixels [30–33]. After completing the aforementioned registration, differential interferometry is performed using a single master image approach, which is followed by MT-InSAR processing. Finally, the data are imported into Stanford Method for Persistent Scatterers (StaMPS) for time-series analysis.

To acquire robust high-coherence points, this paper utilizes the second-generation MT-InSAR technology, where Distributed Scatterers (DS) refer to scatterers with relatively stable and consistent material properties on the ground surface; these exhibit specific characterization features and signal attributes. In the second-generation MT-InSAR technol-

ogy's algorithmic framework, the identification of Statistically Homogeneous Pixels (SHP) is a crucial step. Leveraging the merits of the Kolmogorov–Smirnov (KS) test, this study employs the KS test to identify SHP. During this process, appropriate thresholds are set for SHP selection based on the amplitude difference between two pixels, thus determining the SHP sample set for covariance matrix estimation. Subsequently, phase optimization is performed through Phase Linking (PL), and time-series deformation is analyzed in conjunction with PS and DS [11].

In this study, phase optimization is performed using the Phase Triangulation algorithm [34]. During interferometric processing, among N SAR images, one is chosen as the master image, while the others are suitably resampled, which can be represented as:

$$\mathbf{d}(P) = [d_1(p), d_2(p), \ldots, d_N(p)]^T \tag{1}$$

where $P$ represents any pixel in the image, and $d_i(p)$ denotes the complex reflectance value of pixel $P$ in the $i$th image. Given two data vectors $d(p_1)$ and $d(p_2)$, under a specified significance level, if the hypothesis that $p_1$ and $p_2$ both belong to the same probability distribution function holds, then the pixels $p_1$ and $p_2$ in the two images are statistically homogeneous. We define a set of SHP as $\Omega$, where $\Omega$ conforms to matrix $\mathbf{Z}$, $\mathbf{Z} \in \mathbb{R}^{n \times n}$, $n$ represents the temporal domain, and $l$ represents the spatial domain. Building upon the Central Limit Theorem, the matrix $\mathbf{Z}$ can be represented by a multi-dimensional complex cyclic Gaussian distribution function that obeys zero mean. Therefore, for a comprehensive statistical characterization in the DS framework, the covariance matrix of the identified SHP samples, or its normalized coherence matrix $\mathbf{C}$, can be estimated by the equation:

$$\mathbf{C} = \frac{\mathbf{z}\mathbf{z}^H}{\sqrt{\|\mathbf{z}\|^2 (\|\mathbf{z}\|^2)^T}} \tag{2}$$

The coherence matrix $\mathbf{C}$ can be decomposed into two complex diagonal matrices and one full-rank real symmetric matrix:

$$\boldsymbol{\Psi} = \mathrm{diag}[\boldsymbol{\Psi}] = \mathrm{diag}[\exp(j\phi)] \tag{3}$$

$$\Sigma(\mathrm{P}) = \boldsymbol{\Psi}\boldsymbol{\Gamma}\boldsymbol{\Psi}^H \tag{4}$$

The probability distribution function of SHP is given by:

$$f(z) = \frac{1}{\sqrt{2\pi^n \det(\boldsymbol{\Sigma}_{DS})}} \exp\left\{ -\frac{1}{2}\|\mathbf{z}\|^2_{\boldsymbol{\Sigma}_{DS}^{-1}} \right\} \tag{5}$$

Its second-order matrix can be expressed as:

$$\boldsymbol{\Sigma}_{DS} = \boldsymbol{\Psi}\hat{\boldsymbol{\Gamma}}\boldsymbol{\Psi}^H \tag{6}$$

The maximum likelihood estimation value of the true phase is obtained as:

$$\hat{\Phi} = \mathrm{argmin}_{\Phi}\left\{ \boldsymbol{\Psi}^H \left( \boldsymbol{\Gamma}^{-1} \circ \mathbf{C} \right) \boldsymbol{\Psi} \right\} \tag{7}$$

In the aforementioned equation, $H$ denotes the Hermitian conjugate. $\boldsymbol{\Psi}$ is an N × N complex diagonal matrix, the elements of which represent the true phase values of pixel P, $\Sigma$ signifies the complex coherence model of the SCM. $\boldsymbol{\Gamma}$ is an N × N full-rank real symmetric matrix, $\boldsymbol{\Gamma} \in \mathbb{R}^{n \times n}$, and its elements represent the coherence values across all interferograms. Finally, joint PS and DS processing of the interferometric phases yields the study region time-series displacements, and the resulting time-series data are further processed to obtain the training dataset for the integrated model.

### *3.2. Ensemble Learning Data Preprocessing*

3.2.1. Acquiring Target Variables Based on Distance Features

The MT-InSAR temporal data are processed to harvest the target variables and features requisite for the ensemble learning dataset. For the target variables, based on Spatial Auto-correlation (SAC) theory, high-coherence points situated on the fault zone are segregated, and within a surrounding expanse of 30 m on either flank of the fault line, the mean displacement of all high-coherence points within the identical period is adopted as the target variable [35,36]. As aforestated, the original data are bifurcated into six segments in line with the temporal baseline. During post-processing, each segment yields disparate position coordinates for the high-coherence points. Hence, this paper selects any segment result $P_0$ from the aforementioned six segments and extracts high-coherence points obtained post spatial autocorrelation processing, such as point $a$. Utilizing point $a$ as a reference, and predicated on distance features, the Euclidean Distance (ED) is employed to compute the distance between the reference point and points in segments P1 to P5. Upon appraising the dissimilarity, points with lesser dissimilarity are chosen, and all points in $P_0$ are iteratively traversed [37]. Through this methodology, we can obtain high-coherence points spatially nearly continuous with the reference segment, as depicted in parts t1 to t5 in Figure 3. We calculate the mean displacement of these points within each period for every segment and employ these outcomes as target variables.

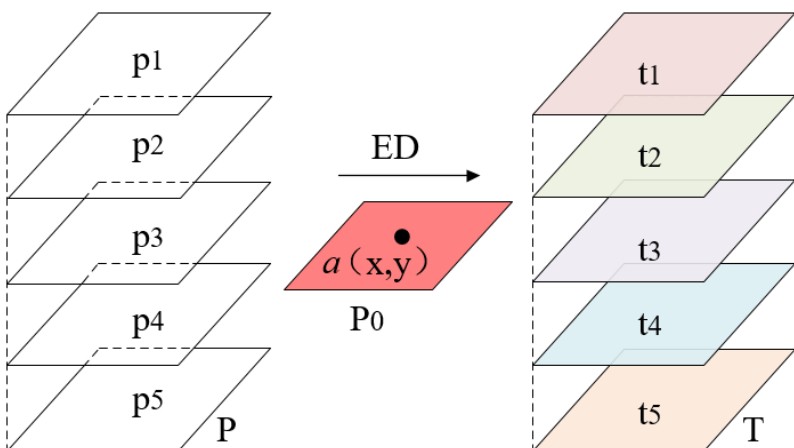

**Figure 3.** Target variable acquisition based on distance features.

3.2.2. Feature Selection Using Spatial Stratified Sampling and Mutual Information

Based on the aforementioned range selection of high-coherence points for target variables, the range for feature data is ascertained (along the fault line from 30 to 260 m), and this is contemplated to adopt the displacement of high-coherence points associated with the target variable within this range as dataset features. Should all points be harnessed for model training, the augmentation to feature dimensions would lead to excessive complexity and overfitting of the model. To pare down the feature dimension whilst ensuring that the sampled points retain the spatial features of the original sample space to the maximal extent feasible, Spatial Stratified Sampling (SSS) is employed to sample the high-coherence points within the defined range [38]. This paper utilizes the coordinates of high-coherence points as spatial variables, partitioning the research area into P × P grids of equal size, with the grid size determined by the quantity of K-means clustering [39], delineated as follows:

$$J = \sum_{i=1}^{n} \sum_{k=1}^{K} r_{ik} \| x_i - u_k \|^2 \tag{8}$$

where $n$ is the number of samples, $K$ is the number of clusters, $u_k$ is the center of mass, and in gridding, $P = K$. Finally, one sample is randomly drawn in each grid, unless there are no coordinate points in this grid.

After ensuring the representativeness of the sample space, we further optimized the high-coherence points with weaker relevance to the target variable, as illustrated in Figure 4. The data obtained through MT-InSAR technology exhibited the nonlinear characteristics of surface deformation. Besides leveraging this nonlinear relationship, the model establishment also necessitated the consideration of associations among features [16]. During the feature selection phase, we aimed to comprehensively evaluate both the nonlinear relationships between features and the target variable and the associations among features. To achieve this goal, we employed the Mutual Information (MI) method. This method, when assessing the relationship between two random variables, can concurrently account for the nonlinear relationship and association between them. Given two random variables X and Y, the mutual information is defined as follows [40,41]:

$$I(X;Y) = \sum_{x \in \mathcal{X}} \sum_{y \in \mathcal{Y}} p(x,y) \log \left( \frac{p(x,y)}{p(x)p(y)} \right) \tag{9}$$

where $p(x,y)$ represents the joint probability distribution of the random variables, while $p(x)$ and $p(y)$ denote the marginal probability distributions of X and Y, respectively.

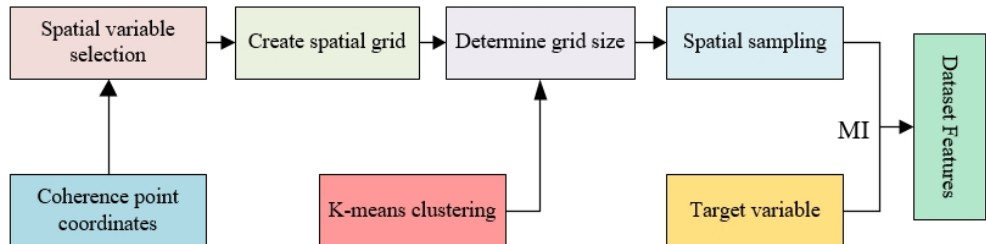

**Figure 4.** Feature selection based on spatial stratified sampling and mutual information.

### 3.3. Model Establishment and Optimization

The Bagging model primarily enhances the model's generalization capability and stability by aggregating the prediction results of multiple weak estimators. Its core idea revolves around using bootstrapping to draw multiple sub-samples from the original training set and training a weak estimator on each sub-sample [42,43]. Predicting fault displacement in the study area falls under regression problems; hence, an average aggregation of the prediction results from weak estimators is conducted:

$$D_i = \{(x_1, y_1), (x_i, y_i), \dots, (x_n, y_n)\} \tag{10}$$

$$\hat{y} = \frac{1}{M} \sum_{i=1}^{M} \hat{y}_i \tag{11}$$

Here, $D_i$ represents the $i$th bootstrap sample and $n$ is the sample size; $M$ denotes the number of weak estimators, and $\hat{y}_i$ is the prediction result from the $i$th weak estimator.

Bagging can enhance the generalization capability and stability of the model, yet the performance of Bagging is significantly influenced by the selection of its hyperparameters. Moreover, traditional grid search and random search exhibit poor efficiency and struggle to thoroughly explore the hyperparameter space. To address this issue, we employed the TPE optimization algorithm to ascertain the optimal hyperparameters for Bagging. TPE is a global optimization algorithm based on a probabilistic model, which constructs a probability model to estimate the conditional probability of hyperparameters and utilizes this model to guide the search for hyperparameters, thereby achieving global optimization of model performance [44].

Employing TPE optimization, we initially delineate the objective function and the parameter space; then, we initialize observation samples by selecting a threshold and bifurcate the historical observations of the objective function into two cohorts: samples

with superior objective function values are categorized into the *l*-group, whilst the lesser ones are slotted into the *g*-group. Presuming the threshold is *t*, the observations are apportioned as follows:

$$X_l = \{x|y(x) < t\} \tag{12}$$

$$X_g = \{x|y(x) \geq t\} \tag{13}$$

Subsequently, for each parameter $x_i$, probability density functions $l(x_i)$ and $g(x_i)$ are constructed for the *l*-group and *g*-group, respectively. The functions can be expressed as follows:

$$l(x_i) = \frac{1}{n_l} \sum_{x \in l} k(x - x_i) \tag{14}$$

$$g(x_i) = \frac{1}{n_g} \sum_{x \in g} k(x - x_i) \tag{15}$$

$$k(u) = \frac{1}{\sqrt{2\pi}\sigma} e^{-\frac{u^2}{2\sigma^2}} \tag{16}$$

For each parameter $x_i$, the Expected Improvement (EI) is calculated as:

$$EI(x_i) = \frac{l(x_i)}{g(x_i)} \tag{17}$$

Ultimately, the parameter values that maximize the expected improvement are chosen to serve as new sample points. The value of the objective function at these new sample points is computed and subsequently incorporated into the observed sample set. This process is iteratively repeated until a sample point yielding the minimum objective function value emerges as the optimal solution.

## 4. Results

### 4.1. MT-InSAR Monitoring Results in the Mining Area

Drawing upon the surface deformation monitoring data, the maximum deformation rate within the mining area is delineated as ranging between 37 to 63 mm/year, as illustrated in Figure 5a. Particularly within the coal mining region, the surface deformation manifests with significant prominence, as demonstrated within the dashed box of Figure 5a. It merits attention that the fault within this mining area traverses the mine boundary along the stratigraphic trend. Despite the presence of protective coal pillars in the fault zone that have not been mined, the monitoring data still reveal noticeable deformation in the proximity. To validate the reliability of the monitoring results, we undertook field inspections and ascertained that the structures within the fault zone were subjected to varying degrees of damage, as depicted in Figure 5b,c. The two rectangular frames near the fault in Figure 5a correspond to the locations shown in Figure 5b,c, respectively. These observations further corroborate that the vicinity of the fault zone indeed experienced deformation, thereby augmenting the support for the hypothesis of fault slippage. The ensuing sections delve into the analysis of the causal relationship between this fault slippage and coal mining activities.

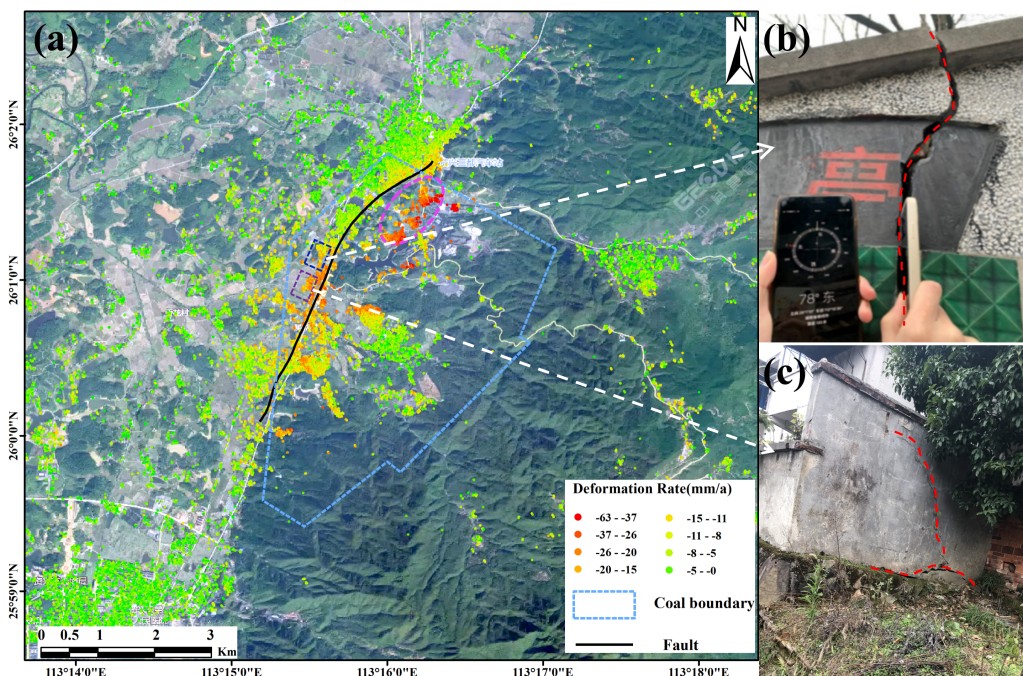

**Figure 5.** Mining area deformation rate and damaged buildings. (**a**) Deformation rate in the mining area (**b**). (**c**) Damaged buildings in the mining area.

### 4.2. Causal Relationship Analysis between Fault Slippage and Coal Mining Activities

As previously mentioned, both the fault zone and the coal mining area exhibit noticeable deformation on the surface. To further investigate the causes of the fault slippage, we employed the Granger causality test and Johansen cointegration test on the monitoring data from the fault and mining areas, providing a comprehensive analysis of this issue from both short-term and long-term perspectives [45]. As illustrated in Figure 6, we selected four representative areas on the fault line: F1, F2, F3, and F4, and processed the displacement time-series data for these areas as described in Section 3.2.1. In the coal mining area, we similarly selected four areas corresponding to F1, F2, F3, and F4—namely M1, M2, M3, and M4 (referred to as 'M areas')—and conducted analogous data processing.

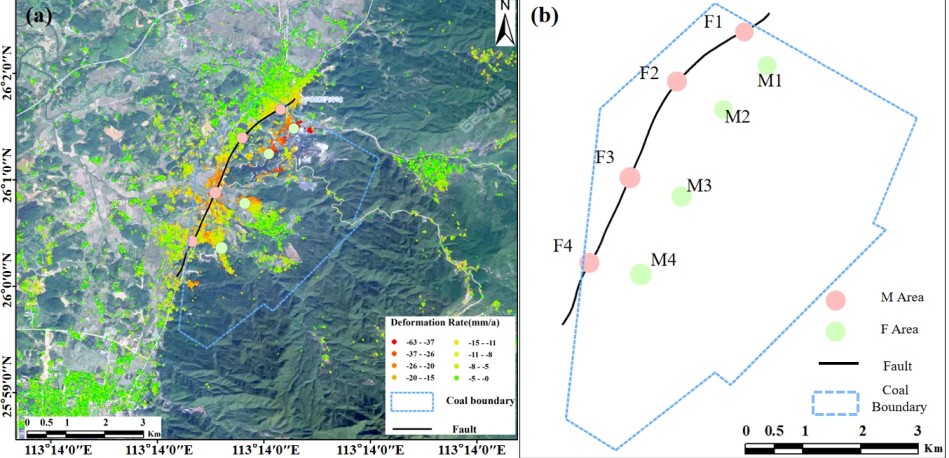

**Figure 6.** Analysis areas of fault zone and mining area surface displacement. (**a**) Causal relationship analysis areas. (**b**) Extraction of causal relationship analysis areas.

Before embarking on the causality analysis, autocorrelation analyses were individually conducted on the displacement time-series data of the F areas and M areas to gauge their stationarity, as clearly illustrated in Figure 7. The autocorrelation function plots

evinced that the autocorrelation coefficients for areas M1 through M4 and F1 through F4 swiftly converged towards zero as the lag steps amplified. This result substantiates that the displacement time-series data for both the F and M areas are stationary, thereby obviating the need for additional processing in subsequent causality tests. After this, a cross-correlation analysis was undertaken on the displacement time-series of the F and M areas, and the range of lag values for the Granger causality test was ascertained to span from 0 to 52.

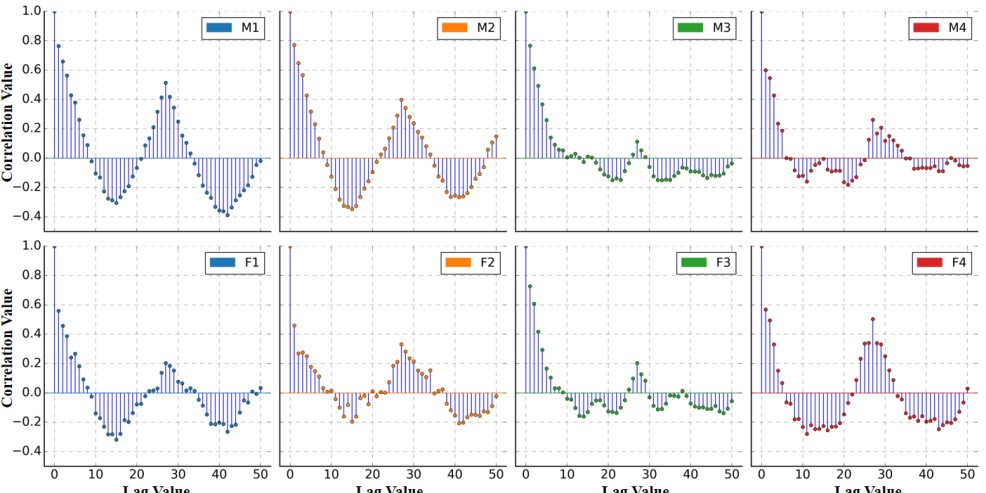

**Figure 7.** Autocorrelation analysis of time-series data.

Building upon prior analyses, both Granger causality and Johansen cointegration tests were executed on the time-series displacement data for regions F and M, the outcomes of which are delineated in Figures 8 and 9, respectively. Within the framework of the Granger causality assessment, the graphical representations substantiate that, at a 95% confidence interval, all but the M–F1 dataset exhibit statistically significant lag values. These revelations suggest that under particular lag conditions, a noteworthy Granger causal linkage exists between the displacements observed in the coal mining sector and the adjacent fault zone. This affirms the proposition that coal extraction activities exert a short-term influence on the stability of the fault zone.

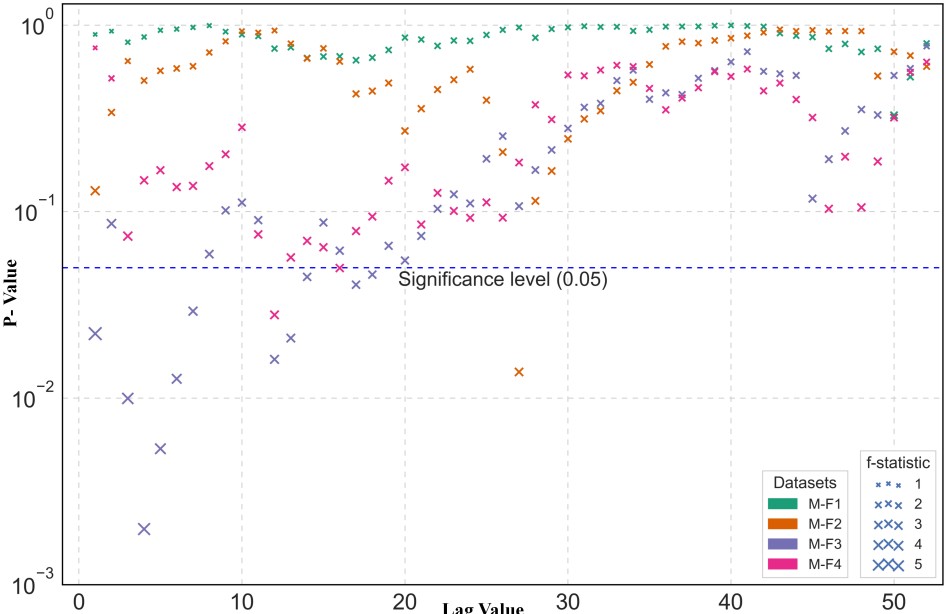

**Figure 8.** Granger causality test.

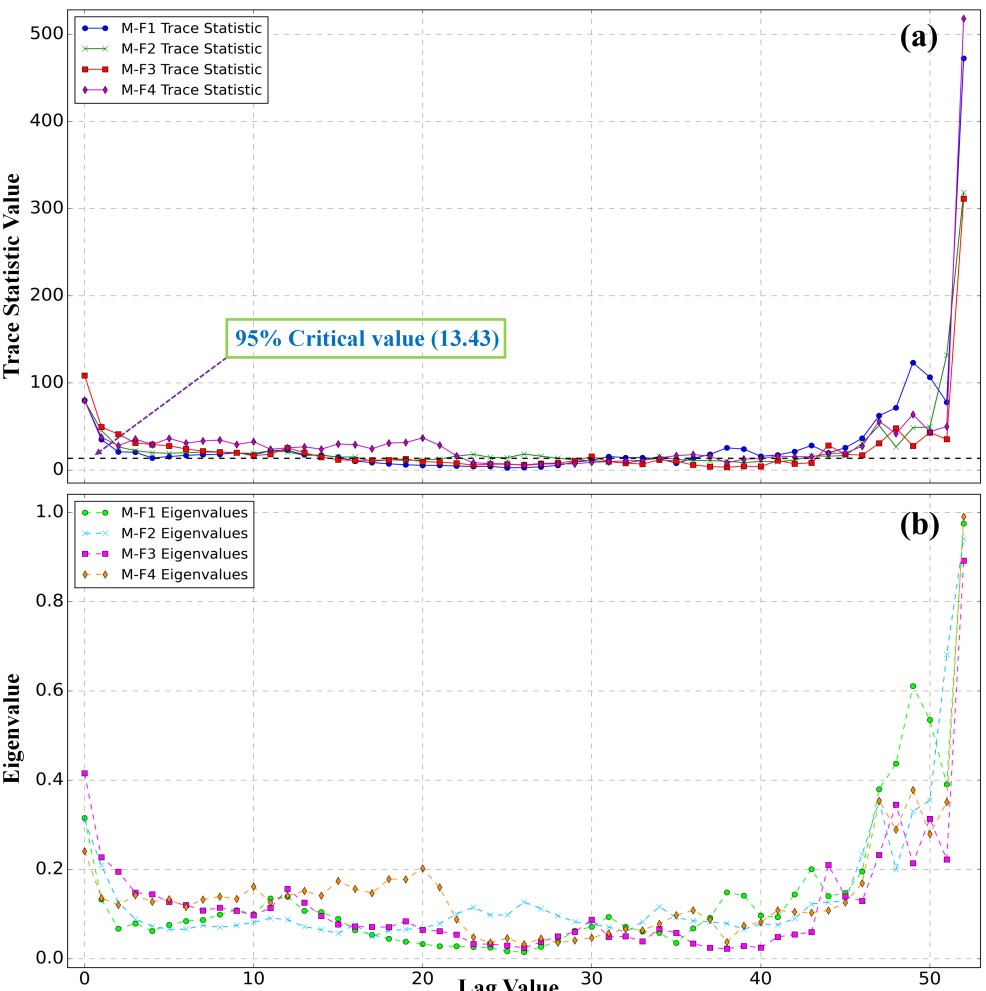

**Figure 9.** Johansen cointegration test. (**a**) Trace statistic value. (**b**) Eiggenvalue.

Within the Johansen cointegration examination, the Eigenvalues pertaining to the four datasets manifest discrepancies under varying lag values but sustain a relatively elevated stature, denoting a more robust long-term cointegration association between the displacement time-series of regions F and M. Additionally, the test statistics for a majority of lag values eclipse the critical threshold at the 95% confidence level (13.43), thereby enabling the rejection of the null hypothesis—the nonexistence of a cointegration relationship. This further validates the cointegration relationship amid the displacement time-series of regions F and M, elucidating the enduring repercussions of coal mining activities on the stability of the fault lines.

Synthesizing the analyses presented above, the following conclusions can be drawn: the mining activities in the coal mine area have significantly influenced the dynamic behavior of the nearby fault in the short term and have also established a long-term stable relationship with it. This finding emphasizes the importance of including displacement data from the mining area as features in the training dataset when developing a model to predict fault slippage.

### 4.3. Establishment and Selection of Fault Displacement Forecasting Model

#### 4.3.1. Model Establishment and Evaluation Scheme

This study, predicated on MT-InSAR monitoring results and the causal relationship between regions F and M, carried out data preprocessing in accordance with the strategy delineated in Section 3.2, thereby constructing an ensemble learning dataset. The target variable of the dataset is the temporal displacement of the mining area fault, as depicted

in Figure 10a, while the feature data consist of the processed temporal displacements of high-coherence points within regions F and M.

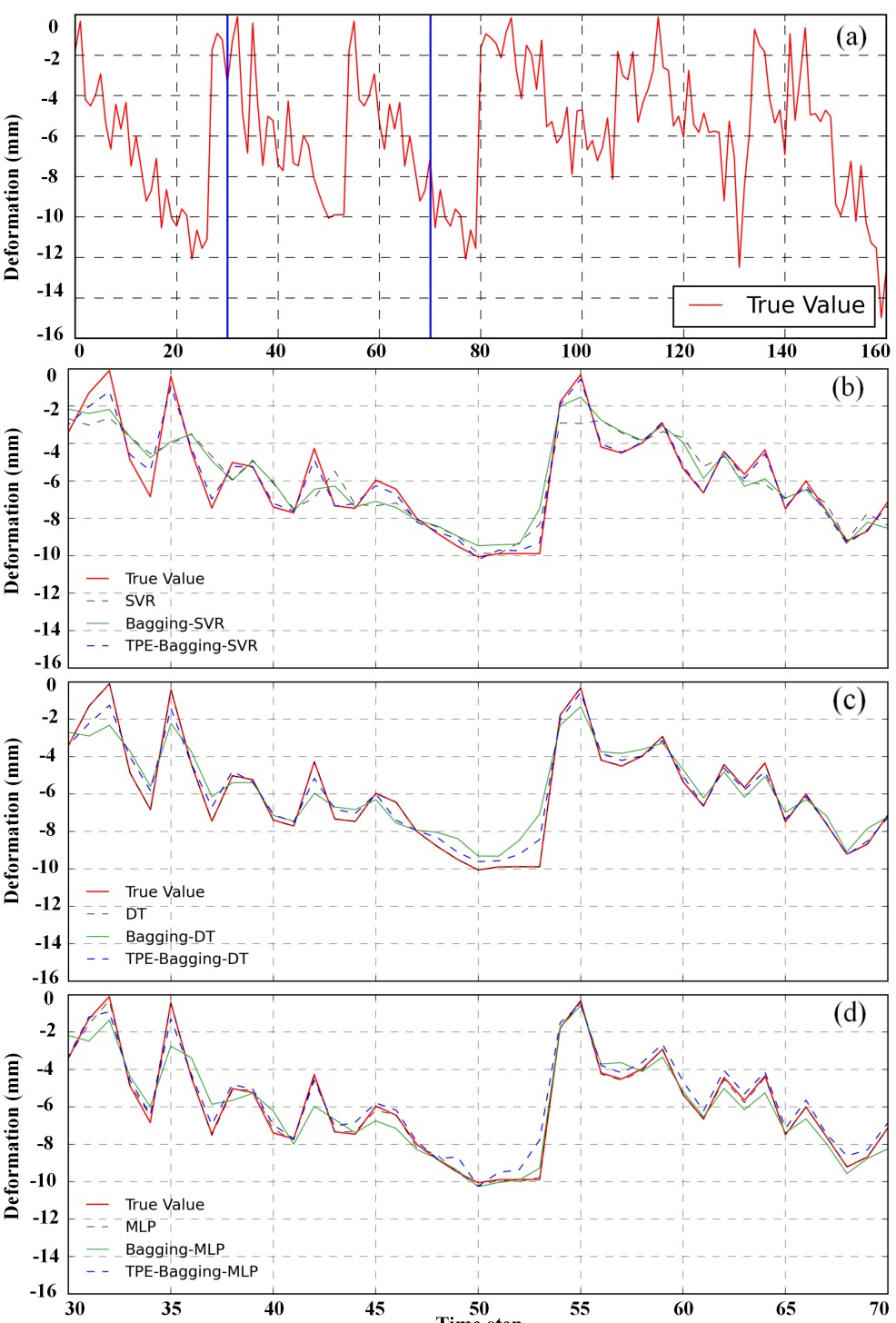

**Figure 10.** Mining fault model prediction results. (**a**) Dataset target variable. (**b**) SVR experiment results. (**c**) DT experiment results. (**d**) MLP experiment results.

Given the regression-centric nature of the research quandary, this paper earmarks SVR (Support Vector Regression), DT (Decision Tree), and MLP (Multilayer Perceptron) as the models for prognosticating fault displacement, and we conduct experimental evaluation predicated on the ensemble learning dataset. The evaluation metrics encompass Mean Square Error (MSE), Root Mean Square Error (RMSE), and Mean Absolute Error (MAE),

with the model performance assessment chiefly anchored on the RMSE of the data test set. Moreover, the Bagging method was harnessed to individually ensemble these three models, culminating in the Bagging-SVR, Bagging-DT, and Bagging-MLP models. The three ensemble models were separately optimized utilizing TPE, yielding the optimized models christened TPE-Bagging-SVR, TPE-Bagging-DT, and TPE-Bagging-MLP. Subsequently, experimental comparisons were orchestrated among the basic models, ensemble models, and optimized models. To furnish a more comprehensive appraisal of model performance, Random Forest (RF) was selected as the baseline model and was juxtaposed with the optimum models employing SVR, DT, and MLP as basic estimators to ascertain the most suitable model for prognosticating fault displacement in mining areas.

### 4.3.2. Model Experimental Analysis

To meticulously assess the performance of the SVR, DT, and MLP models for forecasting fault displacement within the mining area, this section delves into the analysis of these initial models, the Bagging ensemble models, as well as the TPE-optimized ensemble models. The experiment outcomes are lucidly illustrated in Figures 10 and 11, alongside Table 1. Given the manually adjusted hyperparameters in the initial models, the experiments unveil that these initial models yielded suboptimal results, manifesting as either overfitting or underfitting. In Figure 10a, a segment of actual value fitting is captured, as depicted in Figure 10b–d, wherein all points exhibit that the curve predicted by the initial model nearly overlaps with the real curve. In conjunction with the data from Table 1, it can be inferred that overfitting is prevalent in the initial models, whereas Figure 11 distinctly showcases underfitting in the initial models. Nonetheless, the experimental results unanimously indicate that the ensemble and optimization methodologies have bolstered the performance of all three models.

Specifically, the Bagging method can effectively reduce the prediction error of the models, while TPE optimization further improves the prediction accuracy. Compared to the original models, the performance of the ensemble models improved by 10.8%, 28.3%, and 16.5%, respectively. Furthermore, compared to the baseline model, as shown in Table 2, the performance of TPE-Bagging-SVR, TPE-Bagging-DT, and TPE-Bagging-MLP improved by 4.72%, 12.19%, and 25.88%, respectively. After TPE optimization, the performance of these models further improved, with increments of 7.3%, 2.7%, and 25.1%, respectively.

Among all models, the TPE-Bagging-MLP model emerged as the superior performer, exhibiting the lowest RMSE, MAE, and MSE metrics on the test dataset. This underscores that the TPE-Bagging-MLP model is the most apt choice for predicting mine fault displacement. Through the amalgamation of ensemble and optimization techniques, we have not only corroborated the effectiveness of the preprocessing method delineated in Section 3.2 but also showcased that ensemble learning approaches markedly augment the predictive accuracy of MT-InSAR monitoring results. Furthermore, the experimental results also underscore that singular base models such as DT, SVR, and MLP may incur substantial errors and exhibit diminished stability when tasked with predicting mine fault displacement, whilst the employment of the Bagging ensemble and TPE optimization can effectively mitigate these issues.

Based on the experimental results, we conclude that ensemble learning methods can effectively handle MT-InSAR monitoring results to predict mine fault displacement, further verifying the feasibility of the preprocessing methods described in Section 3.2. Concurrently, utilizing single-base models such as DT, SVR, and MLP for prediction can result in substantial errors and poorer stability, while employing the Bagging ensemble and leveraging TPE optimization can significantly enhance prediction accuracy.

**Table 1.** Experimental results of models.

| Model | Train Datasets | | | Test Datasets | | |
|---|---|---|---|---|---|---|
| | RMSE | MAE | MSE | RMSE | MAE | MSE |
| SVR | 2.296 | 1.740 | 5.272 | 1.928 | 1.824 | 3.717 |
| Bagging-SVR | 1.959 | 1.516 | 3.838 | 1.719 | 1.619 | 2.955 |
| TPE-Bagging-SVR | 1.642 | 1.211 | 2.696 | 1.594 | 1.454 | 2.541 |
| DT | 2.213 | 1.492 | 4.897 | 1.903 | 1.818 | 3.621 |
| Bagging-DT | 1.722 | 1.267 | 2.965 | 1.364 | 1.278 | 1.860 |
| TPE-Bagging-DT | 1.552 | 1.194 | 2.409 | 1.327 | 1.308 | 1.761 |
| MLP | 1.751 | 1.311 | 3.066 | 1.983 | 1.898 | 3.932 |
| Bagging-MLP | 1.696 | 1.355 | 2.876 | 1.656 | 1.542 | 2.742 |
| TPE-Bagging-MLP | 1.252 | 1.003 | 1.568 | 1.240 | 1.167 | 1.538 |

**Table 2.** Baseline model experiment.

| Model | Train Datasets | | | Test Datasets | | |
|---|---|---|---|---|---|---|
| | RMSE | MAE | MSE | RMSE | MAE | MSE |
| Random Forest | 1.798 | 1.256 | 3.233 | 1.673 | 1.429 | 2.799 |

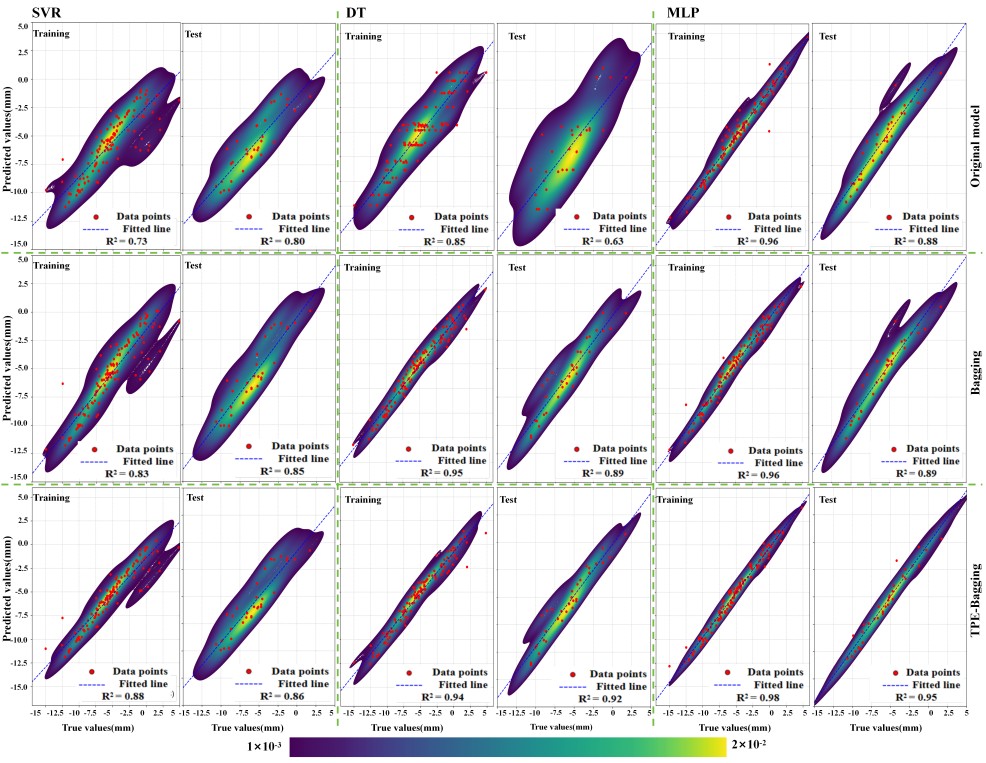

**Figure 11.** Scatter density plot of mining fault model prediction results.

## 5. Discussion

This study conducted a comparative analysis of the performance of basic models (SVR, DT, and MLP), Bagging ensemble models, and their enhancements via TPE optimization for predicting fault slip in mining areas. The results demonstrated that the TPE-Bagging-MLP model excelled across various evaluation metrics. These findings not only validate the effectiveness of the MLP model at handling complex data but also underscore the significance of ensemble learning methods and parameter optimization strategies for enhancing model performance.

The superior performance of the TPE-Bagging-MLP model can be ascribed to the nonlinear modeling capability of MLP and the variance reduction feature of the Bagging

method. The strategic application of ensemble learning has been instrumental for mitigating the instability and overfitting commonly associated with single models. Complementing this, the fine-tuning achieved through TPE optimization significantly bolsters the model's predictive accuracy. This conclusion aligns with existing literature, substantiating the applicability and value of ensemble learning and optimization strategies for forecasting under complex geological conditions. Concurrently, this study emphasizes the necessity of data preprocessing, as described in Section 3.2, highlighting its critical role in geological prediction tasks. This not only provides a foundation for further research into InSAR monitoring data preprocessing but also lays the groundwork for the exploration of more complex models.

Although this study achieved favorable results in experiments, there are still some limitations. For instance, the scale and diversity of the dataset, as well as the preprocessing methods, may impact the model's generalizability. Future research could further address these issues by utilizing more extensive datasets, exploring more efficient foundational models and ensemble methods, and implementing more effective data preprocessing techniques.

## 6. Conclusions

This study addresses the limitations of existing methods for predicting surface deformation in mining areas and the rapid increase in mining monitoring data; we propose a novel framework for surface deformation prediction. The main conclusions are as follows:

(1) By comprehensively employing spatial autocorrelation, distance-based feature methods, spatially stratified sampling, and mutual information methodology, a preprocessing method system tailored for MT-InSAR monitoring results has been formulated. This system facilitates the acquisition of target variables and the selection of features for the ensemble learning dataset, thereby establishing a robust foundation for subsequent analytical endeavors.

(2) Through the meticulous employment of the Granger causality test and Johansen cointegration test, an in-depth unraveling of the causal relationship between fault slip and coal mining activities was executed. For the displacement time-series data of regions F and M, autocorrelation and cross-correlation analyses were conducted, thereby establishing the stationarity of the data and the range of lag values pertinent to the Granger test. The study unveiled a significant Granger causal relationship between coal mining activities and fault slip in the short term, coupled with a long-term cointegration relationship. This alludes to the notion that, transcending short-term impacts, coal mining activities wield a long-term influence on fault slip, thus underscoring the intricate interplay between anthropogenic activities and geological phenomena.

(3) The experimental outcomes elucidate that in comparison to the basic models, the Bagging models exhibited superior performance, and the application of TPE optimization further bolstered their efficacy. Particularly in the Bagging-MLP model, TPE optimization markedly enhanced the performance, whereas the performance augmentations in the optimized Bagging-SVR and Bagging-DT were not as conspicuous. This insinuates that under complex geological contingencies, MLP stands out as more suitable than SVR and DT for employment as a model to prognosticate surface deformation in mining areas. Ultimately, TPE-Bagging-MLP was discerned as the optimal model for predicting surface deformation in mining areas, thus validating the feasibility of the MT-InSAR monitoring data preprocessing methodologies propounded in this paper. Concurrently, this research unfolds a novel and efficacious framework for predicting surface deformation in mining areas, accentuating the potential and the application value of ensemble learning and MT-InSAR technology for the task of predicting surface deformation in mining locales.

**Author Contributions:** Conceptualization, S.L. (Sichun Long) and M.L.; methodology, S.L. (Sichun Long) and M.L.; software, T.L. and W.W.; validation, L.Z., C.Z., and H.D.; formal analysis, M.L.; investigation, M.L. and S.L. (Shide Lu); resources, H.D. and S.L. (Shide Lu); data curation, M.L.; writing—original draft preparation, M.L. and S.L. (Sichun Long); writing—review and editing, C.X.; supervision, T.L. All authors have read and agreed to the published version of the manuscript.

**Funding:** The work was supported by the National Natural Science Foundation of China (grant numbers: 42377453, 42004006); the Hunan Science and Technology Innovation Leading Talents Foundation (grant number: 2021RC4037); the Provincial Natural Science Foundation of Hunan (grant number: 2023JJ30235, 2021JJ40198, 2021-18); and the Hunan Provincial Key Laboratory of Coal Resources Clean-utilization of and Mine Environment Protection Open fund (grant number: E22101).

**Data Availability Statement:** The Sentinel-1A data used in the present study were accessed through the Sentinel Scientific Data Hub (https://scihub.copernicus.eu/dhus/ accessed on 24 October 2023), of the Copernicus Open Access Hub.

**Acknowledgments:** The open-source software DORIS (Version 4.0.8) is used for data processing.

**Conflicts of Interest:** Author Hongjun Ding and Shide Lu were employed by the company China Construction Fifth Engineering Division Corp., Ltd., The remaining authors declare that the research was conducted in the absence of any commercial or financial relationships that could be construed as a potential conflict of interest.

## Abbreviations

The following abbreviations are used in this manuscript:

| | |
|---|---|
| MT-InSAR | Multi-Temporal Interferometric Synthetic Aperture Radar |
| SHP | Statistically Homogeneous Pixels |
| KS | Kolmogorov–Smirnov |
| ED | Euclidean Distance |
| SAC | Spatial Autocorrelation |
| TPE | Tree-structured Parzen Estimator |
| MLP | Multilayer Perceptron |
| DT | Decision Tree |
| SVR | Support Vector Regression |
| TOPS | Terrain Observation with Progressive Scans |
| StaMPS | Stanford Method for Persistent Scatterers |
| GACOS | Generic Atmospheric Correction Online Service |
| RMSE | Root Mean Square Error |
| MSE | Mean Square Error |
| MAE | Mean Absolute Error |

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
