# Peer review of "Research on Prediction of Surface Deformation in Mining Areas Based on TPE-Optimized Integrated Models and Multi-Temporal InSAR"

_remotesensing, doi:10.3390/rs15235546_

Round 1

Reviewer 1 Report

Comments and Suggestions for Authors

This paper amalgamates ensemble learning algorithms with InSAR technology, proposing a novel method for forecasting ground deformation in mining areas. It holds potential for ground deformation prediction in these areas, and demonstrates solid technical and scientific foundations. The innovation is apparent in the processing of ensemble learning datasets as well as in the construction of models for predicting ground deformation in mining areas. Overall, this paper merits attention. However, there are some inappropriate mistakes in this article, which need to be improved.

Comment 1. Line 103: It is recommended to change 42cm/a to 42cm/year for easier understanding. Line 105: "2.7 3.5 m" - is this missing a character or word?

Comment 2. Line 119: The full expression for StaMPS is not provided. Line 135: "Tree-structured Parzen Estimator" can be abbreviated as TPE. Line 252, 253: In equations (14) and (15), the subscript of x should be i.

Comment 3. Line 319: For easier identification, it is suggested that Figure 9 should annotate the critical value where the test statistic exceeds the 95% confidence level.

Comment 4. Line 363: For easier reading, it is advisable to adjust the coordinate values and legends in Figure 11, and to distinguish between the models directly.

Comment 5. Section 5: The experimental results show that the model based on MLP  generally outperforms the models based on SVR and DT. The discussion section should provide an appropriate explanation.

Author Response

Dear Professor,

I would like to express my profound gratitude for taking the time to review my manuscript amidst your busy schedule. Your feedback has been invaluable in guiding the improvement of my paper.

I deeply appreciate your specific suggestions, such as marking the critical values in Figure 9 and ensuring clearer distinction between each model in Figure 11. These modifications have indeed significantly enhanced the readability of the article. I also apologize for any inconvenience caused by the language and sentence structure in my initial submission.

I have meticulously considered each of your comments and have made thoughtful revisions accordingly. Your insights have not only improved the manuscript but have also contributed greatly to my development as a researcher.

Thank you once again for your invaluable assistance. Attached, you will find the specific details of the changes I have implemented.

Sincerely,

Maoqi Liu

Comment 1. Line 103: It is recommended to change 42cm/a to 42cm/year for easier understanding. Line 105: "2.7 3.5 m" - is this missing a character or word?

Response: I have corrected the errors in the document, adding the symbol between sections 2.7 and 3.5, which is highlighted with a purple background at Line 103. Additionally, I changed '42cm/a' to '42cm/year,' marked with a green background at Line 101. (The abstract is annotated in purple, while the introduction is marked in light blue, indicating modifications suggested by another expert.)

Comment 2. Line 119: The full expression for StaMPS is not provided. Line 135: "Tree-structured Parzen Estimator" can be abbreviated as TPE. Line 252, 253: In equations (14) and (15), the subscript of x should be i.

Response: I have made the necessary corrections in the document, including the addition of the full name of StaMPS, which is now located at Line 137 (this section is marked in light blue, indicating modifications suggested by another expert). I have removed the term 'Tree-structured Parzen Estimator,' leaving only its abbreviation, with the change made at Line 122. Additionally, I revised equations (14) and (15), marked in blue font, with the changes located at Line 250 and Line 251.

Comment 3. Line 319: For easier identification, it is suggested that Figure 9 should annotate the critical value where the test statistic exceeds the 95% confidence level.

Response: I have revised the figure, marking the critical value of 13.43 in it. The updated figure is located on page 11, highlighted with a green background.

Comment 4. Line 363: For easier reading, it is advisable to adjust the coordinate values and legends in Figure 11, and to distinguish between the models directly.

Response: I have made modifications to Figure 11 to clearly distinguish each model. The revised figure is now located on page 14, highlighted with a green background.

Comment 5. Section 5: The experimental results show that the model based on MLP  generally outperforms the models based on SVR and DT. The discussion section should provide an appropriate explanation.

Response: For this issue, I have rewritten the Discussion section, which also includes an explanation of why the model with MLP as the weak evaluator performs better than the other two models. In the Discussion section, this has been annotated in light blue font, spanning from Line 387 to Line 389.

Reviewer 2 Report

Comments and Suggestions for Authors

The manuscript “Research on prediction of surface deformation in mining areas based on TPE optimized integrated model and multi-temporal InSAR” presented surface deformation forecasting within mining areas. The manuscript is well organized. The models used in the paper were introduced in detail.  Thus, the manuscript can be accepted after minor revision. Detailed comments are attached in pdf file.

Comments on the Quality of English Language

The article uses a large number of obscure words

Author Response

Dear Professor,

I am writing to extend my sincerest gratitude for taking time out of your busy schedule to review my paper. Your constructive feedback has been instrumental in enhancing the quality of my work.

I particularly appreciate your suggestion to divide Figure 9 into two sub-figures, which indeed makes it easier for readers to understand. I apologize for any inconvenience caused by the language or sentence structure in the original manuscript.

I have carefully considered each point of feedback you provided and have made thoughtful revisions accordingly. Your detailed insights have not only improved the manuscript but have also been invaluable to my growth as a researcher.

Once again, thank you for your invaluable assistance. The specific details of the modifications I have made are attached for your reference.

Sincerely,

Maoqi Liu

Dear Professor, I have carefully reviewed and summarized your comments on the manuscript, and have made the following revisions accordingly:

Comment 1. Please avoid using obscure vocabulary. e.g. " amalgamating, surveillance, nexus, erected,juxtaposition etc.”. More commonly used words can replace those vocabulary.

Response: I have made the corresponding modifications in the document, highlighted with a purple background. Specifically, 'amalgamating' has been replaced with 'integrating' at Line 6; 'surveillance' has been replaced with 'monitoring' at Lines 7 and 23; 'nexus' has been replaced with 'relationship' at Line 11; 'erected' has been replaced with 'constructed' at Line 14; and 'juxtaposition' has been replaced with 'comparison' at Line 17.

Comment 2. Revise"manuscript"to"paper".

Response: I have made the necessary modifications at the corresponding place in the document, which is marked in purple at Line 5.

Comment 3. MT-InSAR is the most commonly used abbreviation.

Response: I have replaced all instances of 'MTI' with 'MT-InSAR' in the article, and marked these changes in purple. Due to the numerous occurrences of 'MTI', I have only annotated the abstract section.

Comment 4. Section 1 lacks an introduction to ensemble learning,Please introduce the usage of the Bagging model etc in previous study.

Response: Thank you for your suggestions. I have added an introduction to the Bagging model in the introduction section, along with relevant literature references. The modified text is highlighted in light blue, from Line 68 to Line 72. (The sections marked in green indicate modifications suggested by another expert.)

Comment 5. In the dataset section, just introduce the data, it is not proper to describe the processing workflow of InSAR.

Response: This suggestion is excellent. I have moved the narrative about data processing to Section 3.1, spanning from Line 125 to Line 137, and have highlighted this section in blue. (The parts marked in green reflect modifications suggested by another expert.)

Comment 6. Line103 to line104:Rephase to avoid confusion.

Response: In this part of the article, there was a missing symbol between sections 2.7 and 3.5. I have now added the symbol and highlighted the modification with a purple background, located at Line 103. (The sections marked in green are modifications suggested by another expert.)

Comment 7. Line119: Replace the word "scrutiny" with other words.

Response: The modifications can be seen in Section 3.1, where I have replaced 'scrutiny' with 'analysis' and highlighted this change with a purple background, located at Line 137.

Comment 8. References about DS selection, phase linking etc should be provided!

Response: I apologize for the oversight. I have now added the necessary references in the relevant section. These can be found in Section 3.1, specifically at Lines 148 and 150.

Comment 9. Please provide the coordinates of the fild pictures.

Response: I apologize for the inconvenience, but we did not record the precise coordinates during our survey; we can only indicate approximate locations. I have marked these approximate locations in the figure with lines accompanied by arrows. The positions of the two rectangular frames in Figure 5(a) correspond to Figures 5(b) and 5(c), respectively. The updated figure is located on page 9. (The light red annotated text represents the opinions of another expert.)

Comment 10. Please explain the physical meaning lag values.

Response: I apologize for the lack of clarity; in the interest of brevity, I did not elaborate in the text. The physical significance of 'Lag values' refers to the time interval considered when analyzing the causal relationship between two time series. Specifically, it denotes the number of time points from one time series that are considered in predicting another time series, essentially representing the time delay of one sequence relative to another.

Comment 11. It is hard to distinguish the trace statistic value and eigenvalue, considering separating Figure 9 into two subplots.

Response: Your suggestion was excellent. I have modified Figure 9 to include two subfigures, one for the trace statistic value and the other for the eigenvalue. The updated figure is now located on page 11, highlighted with a green background.

Reviewer 3 Report

Comments and Suggestions for Authors

It was a good study to determine surface deformations in mining areas. A useful study for using multi-temporal INSAR data in mining areas and optimizing the results. The article exhibits a well-crafted and organized structure, with only a few minor corrections suggested below.

-Because I am not a native English speaker, some of the words used in the article seem a bit too complex. More basic expressions could have been used for non-native English readers. For example;

Line 35-Epitomizes

Line 41-garnering

Line 66-ameliorate

Line 73-bifurcated

Line 90-linchpin

Line 91-demarcates

Line 270-vicinity

-Figure 1(b) should be provided as a closer-up map. You have already shown the general boundaries in Figure 1(a). Use an arrow to depict your area more closely in Figure 1(b).

-Figure 5: The area indicated by the dashed purple line in Figure 5(a) is not specified in the legend."

-In Figure 6(b), you have represented the 'fault' feature in red, but it is shown in black on the map. Additionally, it would be more appropriate to add a scale to Figure 6(b).

Author Response

Dear Professor,

I am writing to express my heartfelt gratitude for the time and effort you have dedicated to reviewing my manuscript amidst your busy schedule. Your insightful feedback has been immensely valuable in guiding the improvement of the paper.

In particular, I greatly appreciate your suggestion to replace less common vocabulary with more commonly used terms, which indeed enhances the accessibility and comprehension for the readers. I apologize for any inconvenience caused by my language, sentence structure, or symbolism that may have been initially unclear or complex.

I have carefully considered each of your comments and have made corresponding revisions to the manuscript. Your detailed and constructive critique has not only enriched my work but also contributed significantly to my personal and professional growth.

Once again, I am deeply thankful for your invaluable assistance and guidance. It has been an enlightening and enriching experience. The specific details of the modifications made are attached herewith.

Sincerely,

Maoqi Liu

Dear Professor, I have carefully reviewed and summarized your comments on the manuscript, and have made the following revisions accordingly:

Comment 1. Line 35-Epitomizes; Line 41-garnering; Line 66-ameliorate;Line 73-bifurcated; Line 90-linchpin; Line 91-demarcates; Line 270-vicinity.

Response: I apologize for the inconvenience. I have replaced certain terms with more commonly used words, and highlighted these changes with a light green background. 'Epitomizes' has been changed to 'exemplifies,' with the revision at Line 35; For the word 'garnering' I have rephrased this sentence. I have marked it with light blue text. The location is from Line 40 to Line 42; 'ameliorate' to 'address,' at Line 64; 'bifurcated' to 'divided,' at Line 74; 'linchpin' to 'cornerstone,' at Line 89; 'demarcates' to 'defines,' at Line 90; and 'vicinity' to 'proximity,' at Line 267. (Text annotated in light blue in the article represents the opinions of another expert.)

Comment 2. Figure 1(b) should be provided as a closer-up map. You have already shown the general boundaries in Figure 1(a). Use an arrow to depict your area more closely in Figure 1(b).

Response: Thank you for your input. I apologize for any confusion; this section is intended only to provide an overview of the study area, primarily focusing on the fault information of the region. More detailed maps can be found in Figures 5 and 6.

Comment 3. -Figure 5: The area indicated by the dashed purple line in Figure 5(a) is not specified in the legend."

Response: Thank you for your suggestion. In Figure 5, the purple dashed lines (now changed to magenta) are used to highlight areas with significant surface deformation. I have re-described this part in the text, so there is no need to add a legend for clarification. Additionally, following another reviewer's advice, I have marked the approximate locations of Figures 5(b) and 5(c) in Figure 5(a) using arrows. The descriptions of the line types used for marking are detailed in the text and highlighted in light purple font, from Line 267 to Line 271.

Comment 4. -In Figure 6(b), you have represented the 'fault' feature in red, but it is shown in black on the map. Additionally, it would be more appropriate to add a scale to Figure 6(b).

Response: I apologize for the oversight. Following your suggestion, I have revised Figure 6, and the updated figure is now located on page 9.
